# Mobility-Enabled Edge Server Selection for Multi-User Composite Services

**Wenming Zhang [1], Yiwen Zhang [1], Qilin Wu [2],\* and Kai Peng [3]**

1   School of Computer Science and Technology, Anhui University, Hefei 230601, China
2   School of Information Engineering, Chaohu University, Chaohu 238000, China
3   College of Engineering, Huaqiao University, Quanzhou 362021, China
\*   Correspondence: lingqiw@126.com

**Abstract:** In mobile edge computing, a set of edge servers is geographically deployed near the mobile users such that accessible computing capacities and services can be provided to users with low latency. Due to user's mobility, one fundamental and critical problem in mobile edge computing is how to select edge servers for many mobile users so that the total waiting time is minimized. In this paper, we propose a multi-user waiting time computation model about composite services and show the resource contention of the edge server among mobile users. Then, we introduce a novel and optimal Multi-user Edge server Selection method based on Particle swarm optimization (MESP) in mobile edge computing, which selects edge servers for mobile uses in advance within polynomial time. Extensive simulations on a real-world data-trace show that the MESP algorithm can effectively reduce the total waiting time compared with traditional approaches.

**Keywords:** mobile edge computing; multi-user edge server selection; resource contention; total waiting time

## 1. Introduction

The development of cloud computing and mobile networks has enabled people to access services using their smart mobile devices to address business anytime from anywhere [1]. However, because of the data exchange on wide area network (WAN) between the user and remote cloud server, the long round-trip latency will be obtained in some way inevitably [2]. Long round-trip latencies may downgrade the user experience especially for latency-sensitive applications.

To tackle this issue, mobile edge computing (MEC) has been proposed, and a large number of small-scale servers are placed at the network edge [3,4]. MEC is regarded as a supplement to mobile devices with relatively limited computational and storage capacity [5], which can enable computation offloading [6] and provide services to users. Service providers deploy their services on hired edge servers to serve users [7,8] so that users can directly connect to edge servers to get services via the wireless communication infrastructure at the network edge (e.g., cellular base station and Wi-Fi access point). Therefore, the round-trip latency to access the edge server will be negligible [2,9,10]. Some mobile applications such as face recognition, natural language processing, and interactive gaming are typically resource hungry and demand intensive computation, which can be run on the edge servers.

However, each edge server can only cover a specific geographical area, and the users within its coverage can connect to it. Due to users' mobility, if the user leaves the coverage area with an unfinished service, service migration should be taken into consideration, and the service will be migrated to another server. It may have data of hundreds of megabytes or several gigabytes to transfer between different servers [5], so that significant network performance degradation will result. In many

cases, single mobile service cannot fully satisfy users' requests. Nevertheless, service composition mechanisms can help achieve complex requirements by composing a set of services [11–13]. Many edge servers may need to be selected to deploy the composite services invoked by the user. The times of service migrations will increase with the number of services invoked by the user. Therefore, it is necessary to select edge servers in advance for mobile users to reduce the number of times of service migration.

Meanwhile, each edge server has limited resources [10,14] so that the aggregate workload generated by users on each edge server must not exceed the computing capacity of the edge server at any time. Many users may request services at the same time. If the resources required by all users' invoking services exceed the total edge servers' resources, some of these users must connect to the remote cloud servers to get services. In this case, round-trip latency should be considered, which will seriously affect users' experience. In addition, each invoked service may need different resources, and each edge server's resource is limited. Therefore, how to select edge servers to deploy services for mobile users to reduce round-trip latency is an important issue.

In this paper, we focus on the problem of how to select edge servers for many mobile users in advance to minimize the total waiting time. The waiting time includes data upload and download time, the response time of the service, round-trip latency, and downtime generated by service migration. The data upload and download time are mainly affected by bandwidth resource. Because different edge servers have different bandwidth resources, when the mobile users select different edge servers to connect to, the data upload and download time generated is also different. To elaborate on this issue, we introduce two scenarios in Section 2. Our objective is to minimize the total waiting time, which is difficult to achieve. In the process of edge server selection, We need to consider many factors including the user's location, the speed of users, the coverage of edge servers, the number of users, and so on.

Therefore, in this paper, we design the Multi-user Edge server Selection method based on the Particle swarm optimization (MESP) algorithm to select edge servers for mobile users in order to minimize the total waiting time. The contributions of this research are as follows:

- We formally model the problem of selecting edge servers for multiple users in mobile environments and establish a computation model of total user time consumption.
- We analyze the resource contention among mobile users and design the MESP algorithm to select edge servers in advance for each mobile user in order to minimize all users' total waiting time. We conduct extensive simulation to verify the effectiveness of the proposed algorithm comparing baseline approaches.

The rest of our paper is organized as follows. Section 2 introduces two examples that show the importance of edge server selection. Section 3 presents the system model of our paper, including prerequisite definitions and multi-user mobility-aware time computation model. Section 4 details our approach of selecting edge servers before users' move. Section 5 presents the experimental simulation evaluation and analysis. Section 6 reviews related work. Section 7 presents the conclusion of our work.

## 2. Motivation Scenarios

In this section, we introduce two specific examples, including multi-user edge server selection with a single service and multi-user edge server selection with composite services. We show that different edge server selection processes will lead to different total user waiting time.

### 2.1. Multi-User Edge Server Selection with a Single Service

In this scenario, we assume that there is a mobile path consisting of two path segments, AB and BC. Due to the limited coverage of the edge server, edge server $s_1$ can only cover path segment AB. Edge server $s_2$ can cover path segments AB and BC. Edge server $s_3$ can only cover path segment BC. In the figure, different edge server's coverage areas are represented with different shaded areas. The data transmission rates of $s_1$, $s_2$, and $s_3$ are 30 kb/s, 20 kb/s, and 10 kb/s, respectively, which are

shown in Figure 1. In path segment AB, three mobile users would like to invoke different services (i.e., user $u_1$ wants to invoke service $T_1$, user $u_2$ wants to invoke service $T_2$, and user $u_3$ wants to invoke service $T_3$, respectively) as shown in Figure 1a. After a while, they reach path segment BC, as shown in Figure 1b. With user's moving, if the user connects to the other server with unfinished service, the service should also be migrated to the corresponding server. The three-layer framework for migrating running applications can be used to optimize the downtime, which divides the service running on the edge server into three layers, including the base layer, the application layer, and the instance layer [15]. The instance layer is the running state of an application, such as CPU, register, non-pageable memory, etc. [5]. When a service is migrated, it will check whether the destination edge server has the copy of the needed base layer and application layer to avoid unnecessary data transferring. A more detailed procedure about service migration can be found in [15]. Considering the limited resources of each edge server, each edge server can deploy a small number of services. For the ease of calculation, we assume that each edge server's capacity is set as four units of computing capacity, and the workload of each service is set as two units (which means that each edge server can accommodate two services). Detailed information about the three services is shown in Table 1, where the *RR*, *UDS*, *DDS*, *RT*, and *SD* columns show the Requested Resources, Upload Data Size, Download Data Size, the service Response Time, and the Service Downtime generated by service migration, respectively.

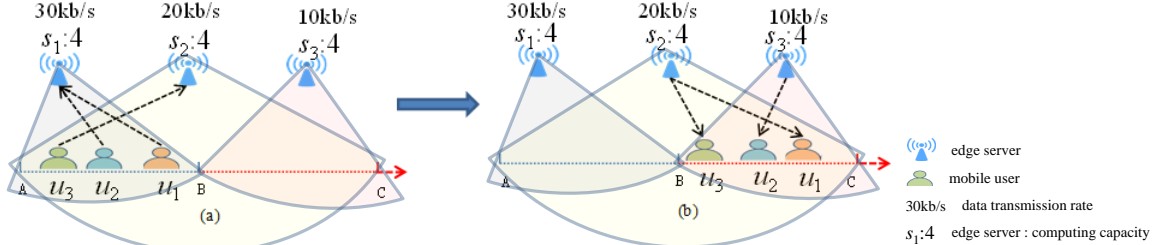

**Figure 1.** The example of multi-user edge server selection with a single service in mobile edge computing (MEC).

Traditionally, because $T_1$ and $T_2$ have a larger upload data size compared with $T_3$, it is intuitive that users $u_1$ and $u_2$ select edge server $s_1$ to deploy the corresponding services, as shown in Figure 1a. At the same time, because the capacity of edge server $s_1$ has been exhausted, user $u_3$ will connect to edge server $s_2$. Over time, three mobile users go into path segment BC and get the response data, as shown in Figure 1b. Since user $u_1$ first goes into path segment BC and is outside the coverage of edge server $s_1$, $u_1$ will connect to the edge server $s_2$, which has a larger data transmission rate compared with $s_3$. Then, user $u_2$ establishes a connection with edge server $s_3$. Thus, the total waiting time of invoking three services is (120/30+300/20+10+2) + (120/30+800/10+10+2) + (40/20+300/20+10) = 154 s. However, if user $u_1$ selects edge server $s_3$, user $u_2$ can connect to edge server $s_2$. In this case, the total waiting time of the three services is (120/30+300/10+10+2) + (120/30+800/20+10+2) + (40/20+300/10) = 129 s, which means that less total waiting time is obtained and user experience is improved. The detailed total waiting time computation model is shown in Section 3.

**Table 1.** Detailed information of the mobile services. The *RR*, *UDS*, *DDS*, *RT*, and *SD* columns show the Requested Resources, Upload Data Size, Download Data Size, the service Response Time, and the Service Downtime generated by service migration, respectively.

| Service | RR | UDS (kb) | DDS (kb) | RT (s) | SD (s) |
|---------|----|----|----|----|----|
| T1 | 2 | 120 | 300 | 10 | 2 |
| T2 | 2 | 120 | 800 | 10 | 2 |
| T3 | 2 | 40 | 300 | 10 | 2 |

### 2.2. Multi-User Edge Server Selection with Composite Services

Figures 2 and 3 illustrate a more complicated example of multi-user edge server selection regarding service composition in the mobile edge computing environment.

In the real world, users always invoke a series of services when they move [16]. As shown in Figure 2, three mobile users would like different composite services in the mobile path. The number in the upper left corner of each service icon indicates the size of the upload data. The number in the upper right corner of each service icon indicates the download data size. The number below each service icon indicates the response time of the service. Suppose that the information of all services, including the upload data size, download data size, the average service response time, and so on, is predetermined. We assume that the workload of each service is two units and the total computing capacity of each edge server is four units. The service downtime generated by the service migration time is set as 2 s [15].

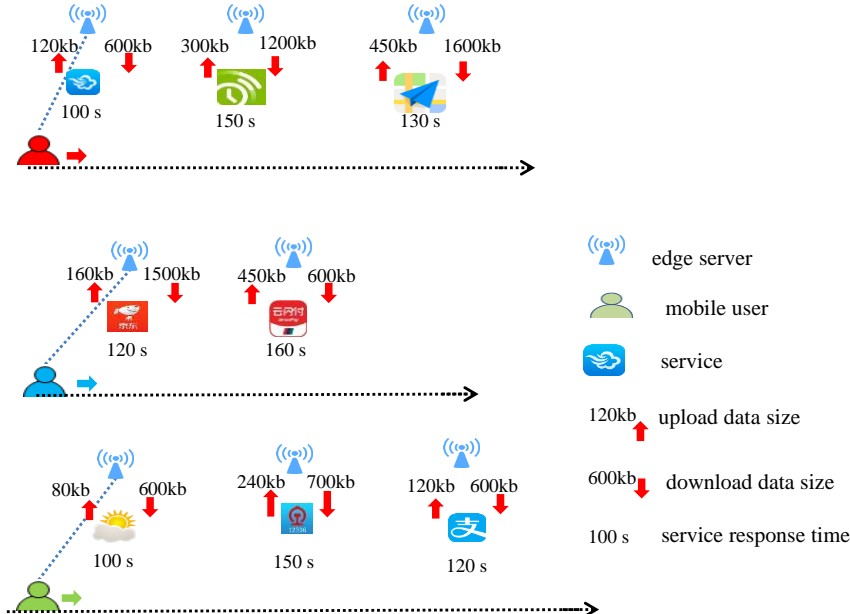

**Figure 2.** Examples of service composition.

The problem is how to select edge servers for mobile users to minimize the total waiting time. The traditional method tries to select the edge server with the most bandwidth when the user wants to invoke the service. The method only minimizes the single-user latency of uploading input data and downloading output data. The data transmission time of the follow-up users cannot be guaranteed due to the limited resources of each edge server and the mobility of the user. As shown in Figure 3a, the total waiting time of all services is 1666 s using the traditional method. However, if the user's mobility and the limited resources of each edge server are taken into consideration when selecting the edge server, a better edge server selection result can be obtained, and the users' waiting time can be reduced. The total waiting time of the all services is 1630.5 s according to the selection process shown in Figure 3b.

Hence, it is important to consider users' mobility and the limited resources of an edge server when selecting edge servers for composite services in order to reduce users' waiting time. The process in which the former user selects edge server may affect the follow-up user's edge server selection, which could affect the total waiting time. Therefore, we need to find a method that selects edge servers for mobile users in advance to minimize users' total waiting time.

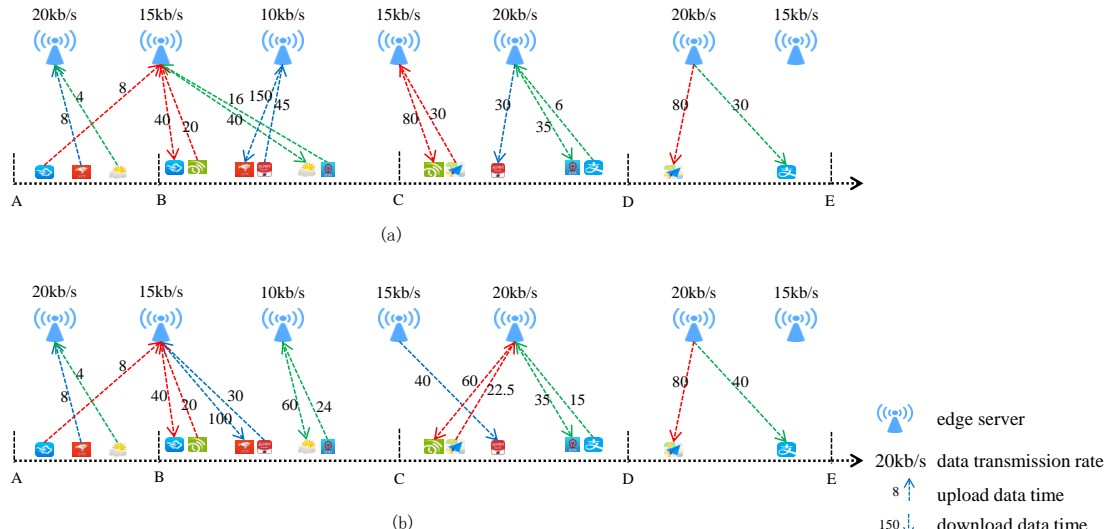

**Figure 3.** The examples of multi-user edge server selection with service composition in MEC.

## 3. System Model

In this section, we first give some clear definitions of the key concepts in the scope about multi-user edge server selection in mobile edge computing. Then, the computation model of single-user time consumption is presented with the known path. Finally, we model how to compute the multi-user total waiting time in the scenario of MEC.

### 3.1. Prerequisite Definitions

The basic concepts including the definitions of the mobile service, the edge server, the mobile path, and the user's moving are formally introduced.

**Definition 1** (Mobile service). *A mobile service is represented by a four-tuple* $(I, O, C, QoS)$, *where:*

*(1) I is the input parameters;*

*(2) O is the output parameters;*

*(3) C represents the resources required by the service, which is an n-tuple* $< c_1, c_2, \cdots, c_n >$, *where each* $c_i$ *is the resource type, including CPU, RAM, VRAM, etc.;*

*(4) QoS is an n-tuple* $< q_1, q_2, \cdots, q_n >$, *where each* $q_i$ *denotes a QoS property of a service, including execution cost, response time, throughput, reputation, etc.*

In this paper, we only consider one QoS property (i.e., response time). This is because the response time of the service is a part of the total waiting time. Users' mobility only affects the variation of the data transmission bandwidth between users and different edge servers, which will affect the data transmission time. The reason is that due to edge server's limited coverage, if the mobile user leaves the one edge server's coverage and goes into another edge server's coverage, the mobile user must connect to the new edge server to get sources. The data transmission rate between the mobile user and the new edge server may be different from the data transmission rate between the mobile user and the source edge server. The upload data size and download data size can be obtained from $I$ and $O$, respectively.

**Definition 2** (Edge server). *An edge server is represented by a four-tuple* $((x, y), radius, C, r)$, *where:*

*(1)* $(x, y)$ *is the longitude and latitude of the edge server;*

*(2) radius is coverage radius of the edge server;*

(3) *C represents the capacity of an edge server and is an n-tuple $< c_1, c_2, c_i, \cdots, c_n >$, where each $c_i$ is the resource type of an edge server, including CPU, RAM, VRAM, etc.;*

(4) *r is the average data transmission rate between the user and the edge server.*

Edge servers are deployed in a distributed fashion (usually near a cellular base station), and because each edge server only covers a specific geographical area, the proximity constraint should be considered. Only the users located within the coverage of an edge server can connect to the edge server [7]. Meanwhile, each edge server has a limited computing capacity [17] denoted as *C*. Therefore, the aggregate workload generated by services on a server must not exceed the remaining capacity of that server. At the same time, we assume that the remote cloud servers have sufficient computing resources and the user can connect to them anytime from anywhere [18].

**Definition 3** (Mobile path). *The mobile path is modeled as a triple $(P, Se, F)$, where:*

(1) *$P = \{p_i\}_{i=1}^{i=n}$ is the set of discrete location points (the mobile path is composed of lines between two adjacent points);*

(2) *$Se = \{se_i\}_{i=1}^{i=m}$ is a set of discrete path segments of the mobile path (the mobile path is composed of all path segments);*

(3) *F is a mapping function between the set of location points and path segments: $F(p_k, p_{k+1}, \cdots, p_{k+\tau}) \to se_i$*

To avoid blank areas, the coverages of adjacent edge servers usually partially overlap. Therefore, we divide the mobile path into many segments, and each segment $se_i$ is covered by the same edge servers. According to function *F*, the adjacent points covered by the same edge servers are chosen as a set; thereby, all the lines between two adjacent points are connected to form a segment. In the process of edge server selection, when the user is in a path segment, the corresponding candidate edge servers can be selected.

**Definition 4** (User's moving). *The process of user's moving can be denoted by a tuple $(Sp, T, L, G)$, where:*

(1) *Sp denotes the initial location of the user in the mobile path;*

(2) *$T = \{t_i\}_{i=0}^{i=n}$ is a set of discrete time points, with $t_0$ as the start time and $t_n$ as the stop time;*

(3) *L is a set of discrete location points of the user;*

(4) *F is a mapping function between time and location: $\forall t_i \in T, F(t_i) \to L$.*

In Definition 4, when the user is in initial position $Sp$, the time is set as $t_0$; when the user's invoking services are finished, the time is set as $t_n$. In addition, function *F* indicates the speed while the user moves. If the user moves with a high speed in a certain region, the time consumption is small; if the user moves with a low speed in a certain region, the time consumption is big.

*3.2. Multi-User Mobility-Aware Time Latency Computation*

With MEC, services can be housed in edge servers that can be used to accommodate service requests from users located in their coverage regions [19]. In this section, we discuss the time consumption computation model of the users.

**Definition 5** (Server selection). *Given a service $ws = (I, O, C, QoS)$ and mobile path $mp = (P, Se, F)$, suppose that ws is invoked at time $t_1$; then, according to Definition 3, each Se is covered by the same edge servers. When the user is located in the Se, the user can select the corresponding candidate servers that meet Equations (1) and (2).*

$$d(r, (x, y)) \leq radius_r \tag{1}$$

According to Definition 2, the user who is not positioned within the coverage of an edge server will not be able to connect to it. Therefore, the distance between the user and the edge server should be less than the coverage radius of the edge server. Meanwhile, due to the limited resources of the edge server, if there are many users connected to an edge server to request service at the same time, the edge server may be overloaded. When the user connects to an edge server, the capacity constraint has to be taken into consideration:

$$C \leq C_r \tag{2}$$

$C$ and $C_r$ represent the resources required by the service $ws$ and the remaining resources of the edge server, respectively. Under the condition of satisfying Equation (2), if the user connects to an edge server to invoke a service, the capacity of the edge server will decrease (i.e., $C_r = C_r - C$). When the user leaves the coverage radius of the connected edge server, the remaining resources of the edge server will increase (i.e., $C_r = C_r + C$).

Next, we discuss the time consumed by the user in invoking services. The time consumption is mainly decomposed into three parts, namely the time latency of transmitting input data, the time latency of transmitting output data, and the response time. Sometimes, when the user connects to the remote cloud servers, the round-trip latency should be added to the time consumption. The round-trip belongs to the transmitting time. When the user connects to the remote cloud servers, some data firstly are transmitted between the mobile device and the network access point (e.g., base station, Wi-Fi). After that, the data will be transmitted between the network access point and the remote cloud server and passed through multi-hop network nodes, whose transmitting time is called the round-trip. The round-trip latency is calculated only when the user connects to the remote cloud. If the user connects to the edge node, the data will not need to be transmitted between the network access point and the remote cloud server. We only need to calculate the data transmitting time between the mobile device and the edge node. Meanwhile, because of the limited radius of the single edge server, when the user leaves the coverage of an edge server with unfinished service and connects to another server, service downtime generated by service migration should be taken into consideration.

**Definition 6** (Time consumption). *Given a service $ws = (I, O, C, QoS)$ and a selected server $s_i$, suppose that $ws$ is invoked at time $t_1$. The time consumption of invoking the service is given by:*

$$tc = t_{du} + Q_{ws} + t_{dd} + t_{rt}I_1 + t_{dt}I_2 \tag{3}$$

*where:*

(1)    *$t_{du}$ is the time latency of uploading input data, which is given by:*

$$t_{du} = \frac{D(I)}{r_{s_i}} \tag{4}$$

     *where $D(I)$ is the data size of $I$ and $r_{s_i}$ is the data transmission rate between the server $s_i$ and the user $u$.*

(2)    *$Q_{ws}$ is the response time of service $ws$;*

(3)    *$t_{dd}$ is time latency of downloading output data, which is given by:*

$$t_{dd} = \frac{D(O)}{r_{s_j}} \tag{5}$$

     *where $D(O)$ is the data size of $O$ and $r_{s_j}$ is the data transmission rate between the edge server $s_j$ and the user $u$;*

(4)    *$t_{rt}I_1$ denotes the round-trip latency, and $I_1$ is an indicator function, which is expressed as:*

$$I_1 = \begin{cases} 1, & \textit{if the user connects to the remote cloud} \\ 0, & \textit{otherwise} \end{cases} \tag{6}$$

(5) $t_{dt}$ denotes the downtime generated by service migration, and $I_2$ is an indicator function, which is expressed as:

$$I_2 = \begin{cases} 1, & \text{if the service is migrated} \\ 0, & \text{otherwise} \end{cases} \tag{7}$$

The time consumption computation method for invoking a service is presented in Definition 6. However, in the real world, users always invoke a series of services when they are moving [11]. Therefore, the time consumption for which a user invokes the entire service composition can be calculated as:

$$Utc = \psi_{ws \in Sws} tc_{ws} \tag{8}$$

where $Sws$ is the set of composite services and $\psi$ is an operator that integrates the values of time consumption of invoking composite services. The integration rules include $\sum$ representing summation, $\prod$ representing the product, *max* representing the maximum, and *min* representing the minimum. For the ease of calculation, we assume that the composite services are in a sequential execution path, and we only use the $\sum$ integration rule.

Therefore, we can get the multi-user mobility-aware time latency computation as follows:

$$SUtc = \sum_{1}^{U} Utc \tag{9}$$

where $U$ denotes the set of all users.

## 4. Edge Server Selection Method

We can get the total waiting time of mobile users according to Equation (9). However, from Section 3, we can know that selecting edge servers in advance for users is the prerequisite for computing the total waiting time. Therefore, in this section, we study how to select edge servers for many mobile users with a known path.

### 4.1. Resource Contention among Mobile Users

According to Definition 5, the aggregate workload on one edge server cannot exceed the edge server's total capacity. Therefore, if the resources of an edge server have been exhausted, the follow-up users will not select the edge server to invoke resources over a period of time. Next, we analyze the edge server resource contention among multiple users in mobile edge computing.

Although a mobile service's response time will not change when it is installed on different servers, the total waiting time of service invocation will still be changed when the user connects to different servers. The reason is that the data transmission rate is not the same between the user and different edge servers. In addition, the capacity of each edge server, service migration, and other factors will also affect the total waiting time, thereby reducing the user experience. Therefore, it is a crucial matter to decide how to select edge servers for different users.

We assume that a random decision solution is generated for each user selecting servers, and each user can select edge servers to invoke services depending on the random decision solution. The decision solution is guaranteed to be available only when there is only one user in the mobile path or a user does not start to invoke services before the previous user finishes invoking services. However, in the real world, there are always many mobile users in the path. Due to the limited capacity of the edge server, if excessive mobile users select one edge server at the same moment, there must be some users that cannot get the response from the edge server. In this case, users' experience may be degraded, and thus, the random decision solution is not feasible.

Therefore, we design the renewal algorithm to make the random decision solution feasible. The renewal algorithm can make the workload on each edge server less than its capacity when the user

selects edge servers according to the random decision solution. The algorithm begins with initialization (Line 1), which gets a new decision solution by copying the original decision solution. Then, according to the original decision solution $p_1$, all mobile users select the predetermined edge servers to invoke services (Lines 2–5). If a user selects the edge server $s$ and it cannot have the service installed to fulfill the request from user $u$ with exhausted capacity (Line 6), another server $s'$ that has sufficient remaining capacity will be randomly selected for the user (Line 7). Thereby, the information for selecting $s'$ for the user instead of $s$ will be updated into $p_2$ (Line 8). Finally, the feasible decision solution $p_2$ will returned (Line 13).

### 4.2. Multi-User Edge Server Selection Method Based on PSO

The quantity of alternative edge servers for users increases exponentially with the number of edge server increasing. If the enumeration method is used to select edge servers for users, the complexity is $O(km^n)$, where $k$ is the number of users, $m$ denotes the average number of candidate servers for invoking single service, and $n$ denotes the average service number of single-user service composition. The enumeration method will not be practical with the scale of the problem increasing. Thus, a multi-user edge server selection method based on particle swarm optimization is proposed to resolve this problem, called MESP. An approximated optimal decision solution within polynomial time can be obtained by the method.

The PSO algorithm is a population-based stochastic optimization technique inspired by social behavior or bird flocking [20–22]. Meanwhile, it has the following advantages: there are only a few parameters that need to be adjusted, which makes it easy to implement; individual local information and group global information are used to search for the optimal decision solution. Therefore, we propose the multi-user edge server selection method based on the PSO. Next, we show how the MESP algorithm is applied to the edge server selection problem.

In this algorithm, we encode the multi-user server selection process as a particle, and all the particles form a population. The algorithm begins with computing the total waiting time generated by each particle $x_i$, and the current optimal position of each particle is recorded as $xbest_i$ (Lines 2–3). Then, the best swarm position is obtained, and the corresponding total waiting time can be obtained by Equation (9) (Line 5). Next, each particle velocity is updated by Equation (10), and each particle position is updated by Equation (11) (Lines 7–8). After that, if updated particle position $x_i$ is not a feasible decision solution, $x_i$ will be modified by Algorithm 1 (Lines 9–10). Finally, the best swarm position $gbest$ and minimizing the total waiting time $fgbest$ is returned.

$$v_i = vw + c_1 R_1 (xbest_i - x_i) + c_2 R_2 (gbest - x_i) \tag{10}$$

where $R_1$ and $R_2$ are random numbers.

$$x_i = x_i + v_i \tag{11}$$

This proposed approach works well only when it has the known path of the mobile users, which means that we know each user's future mobile path so that we can select edge servers for the user in advance. An alternative way to get mobile users' paths is to make use of prediction methods utilized often in wireless mobile computing and communication. In addition, when the user uses the navigation function, we can also get the user's moving path.

---

**Algorithm 1:** Renewal algorithm.

---

    **Input:** original decision solution $p_1$
    **Output:** renewal decision solution $p_2$
    $p_2 = p_1$
    **while** all services are not finished **do**
        **for** each user $u$ **do**
            $u$ selects edge severs $S$ according to $p_1$
            **for** each selected server $s$ in $S$, **do**
                **if** $s$ cannot meet the requirement from $u$
                    It randomly selects another server $s'$ that has sufficient remaining capacity
                    $p_2 \xleftarrow{s'} p_1$
                **end if**
            **end for**
        **end for**
    **end while**
    **return** $p_2$

---

---

**Algorithm 2:** MESP algorithm.

---

    **Input:** iteration times $it$, constant inertia weight $w$, cognitive and social parameters
           $c_1, c_2$, quantity of particle $xSize$, initial random particle position and velocity $x, v$
    **Output:** best swarm position $gbest$ and minimizing total waiting time $fgbest$
    **while** not stopping
        **for** each particle $x_i$, **do**
            compute response time $f(x_i)$ of each particle $x_i$, and set $xbest_i \leftarrow$ best individual
      particle position
        **end for**
        $gbest \leftarrow$ best swarm position, $fgbest = f(gbest)$
        **for** each particle $i$ **do**
            $v_i \leftarrow$ update particle velocity
            $x_i \leftarrow$ update particle position
            **if** $x_i$ is not feasible
                $x_i$ is modified by Algorithm 1
            **end if**
        **end for**
    **end while**
    **return** $gbest, fgbest$

---

## 5. Simulated Experiments and Analysis

In this section, we evaluate the performance of our approach by extensive experiments with a comparison to two baseline approaches. All the experiments were conducted on a Windows machine equipped with Inter Core i7 (3.6 GHz) and 16 GB RAM. The algorithms were implemented by Python 3.6.

### 5.1. Baseline Approaches

To our best knowledge, MESP is the first attempt to consider edge server selection for many mobile users in MEC for composite services. Due to the issue of user's mobility and the limited capacity of the edge server, existing approaches designed for static users cannot be directly applied to the many mobile users environment. Thus, in the experiments, the MESP algorithm was benchmarked against two baseline approaches for multi-user edge server selection, namely the *random* and *traditional*algorithms:

1. *Random*: Each user will randomly select an edge server as long as the server has sufficient remaining resources to accommodate the invoking service and has the users within its coverage.
2. *Traditional*: Each user will select an edge server with the least data transmission time as long as the server has sufficient remaining resources to accommodate the invoking service and has the user within its coverage.

### 5.2. Experiment Settings

We used the dataset from Shanghai Telecom [23,24] to obtain the location data of edge servers. The coverage radius of each server was set within a range of 450 m–750 m [7]. According to [11], the data transmission rate between users and edge servers was set within a range of 100 kb/s–800 kb/s, and the data transmission rate between users and remote cloud server was 100 kb/s. In addition, for the ease of calculation, we only considered one resource type (i.e., CPU) in this paper, and more complicated scenario (i.e., more than one resource type) can be easily generalized. The CPU capacity of each server was set as a random number, and the range was from 2–6. We assumed that the CPU capacity of the remote cloud server was enough. The user's speed was set within 1 m/s–7 m/s. To get all users' moving paths, we randomly selected two points on the *BaiduMap*and obtained a path between two points by the navigation function. Unless stated otherwise, we assumed that the round-trip latency of remote cloud servers was 200 ms [9], and the downtime generated by the service migration was 2.0 s [15].

### 5.3. Experiment Results and Analysis

In this part, we first present the simulation results on the impact of different number of resources, the number of service sin each composing service, and the number of users. The superiority of the proposed algorithm was verified by comparing our algorithm with two baseline approaches. Then, we analyzed how the round-trip and the service downtime generated by service migration impacted the total waiting time in MEC.

#### 5.3.1. The Impact of Resources

According to Definition 5, because of the limited resources of each edge server, the resources required to invoke a service must be less than the selected edge server's remaining resources. Therefore, we first studied the impact of different edge server's resources on the total waiting time. We set the number of users to 14; the number of services invoked by each user was randomly generated from 4–10, and the number of edge server's resources was varied from 2–6. As shown in Figure 4, we can see that the total waiting time generated by MESP and traditional method decreased with the increasing of the edge server's resources. This was because that each edge server could host more services with the increase of the edge server's resources. When users connect to the remote servers, the round-trip should be taken into consideration. With the increasing of edge server's resources, more users can obtain services from edge servers instead of remote cloud servers. The round-trip latency was reduced, which means the total waiting time was also reduced. For the random algorithm, it selected edge servers randomly. Therefore, with the increasing of edge server's resources, the total waiting time generated by random algorithm may not decrease steadily. From Figure 4, we know that under the same conditions, the MESP algorithm outperformed the baseline approaches.

#### 5.3.2. The Impact of the Number of Services

In this subsection, we vary the number of service invoked by each user to examine the impact on the total waiting time. The user number was set 14, and the number of service invoked by each user was changed from four to 10. Figure 5 illustrates the total waiting time affected by the number of services invoked. We can see that with the increasing of the service number, all users' waiting time generated by the three methods increased almost linearly. The reason was that with increasing number

of services in a single-service composition, there were more services that needed to be invoked. More services mean that more data needed to be transmitted, and a greater response time of the service will be obtained. The response time and data transmission time are the main components of the total waiting time. It is obvious that MESP could obtain the least total waiting time compared with the other two baseline approaches under the same conditions.

### 5.3.3. The Impact of Users

Then, we examined the impact of the number of users on the total waiting time, as shown in Figure 6. The service number was randomly generated from 4–10. From the figure, we can see that the total waiting time generated by the three methods increased with the number of users increasing. On the one hand, because each user will invoke some services, increasing the users will invoke more services. On the other hand, because the resources of the each edge server are limited, with increasing users, some users have to connect to the remote cloud servers to get services, and the round-trip latency needs to be taken into consideration. Therefore, the total waiting time will be increased by increasing the users. From the experiment, we can conclude that our MESP algorithm outperformed the traditional and random methods.

From the experiments above, we can conclude that under the same conditions, MESP always outperformed the baseline algorithms. Next, we examine the impact of round-trip latency and service downtime on the total waiting time using the MESP algorithm.

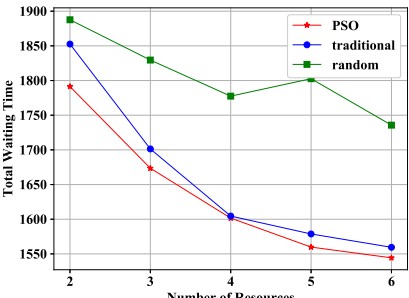

**Figure 4.** The total waiting time with a different number of resources.

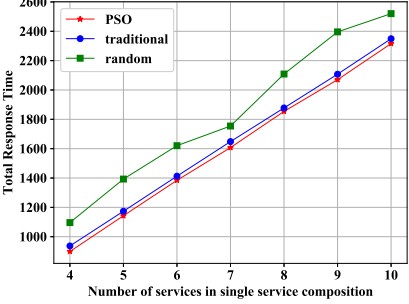

**Figure 5.** The total waiting time with a different number of services in a single-service composition.

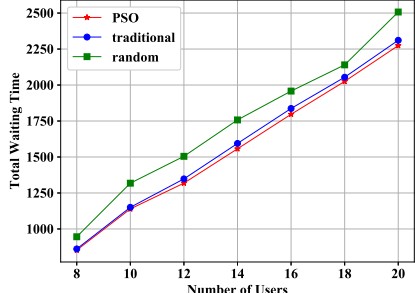

**Figure 6.** The total waiting time with a different number of users.

### 5.3.4. The Impact of the Round-Trip

When the user selects remote cloud servers to deploy services, the round-trip latency between the user and remote cloud servers should be taken into consideration. In Figure 7, we show how edge servers' resources, services' number and users' number impacted the total waiting time using the MESP algorithm. From Figure 7a, we can see that with the edge server's resources increasing, the total waiting time decreased. Under the same conditions, less round-trip latency would result in less total waiting time. When the resources of the edge server were small, it would make many users connect to the remote cloud servers, so that the impact of the round-trip latency was obvious. With the edge servers' resources increasing, more users will connect to the edge servers, so that the impact of the round-trip latency will be less obvious.

From Figure 7b, we can see that with the increasing number of services invoked by each user, the total waiting time also increased. Because the edge server's resources and users' number stayed unchanged, the users connected to remote cloud servers would not change. Therefore, the effect of different round-trip latencies was not obvious.

As shown in Figure 7c, the total waiting time increased with the increasing user number. Because the edge server's resources did not change, with increasing the user number, more users would connect to the remote cloud server. Therefore, the impact of different round-trip latency was more and more obvious with the number of users increasing.

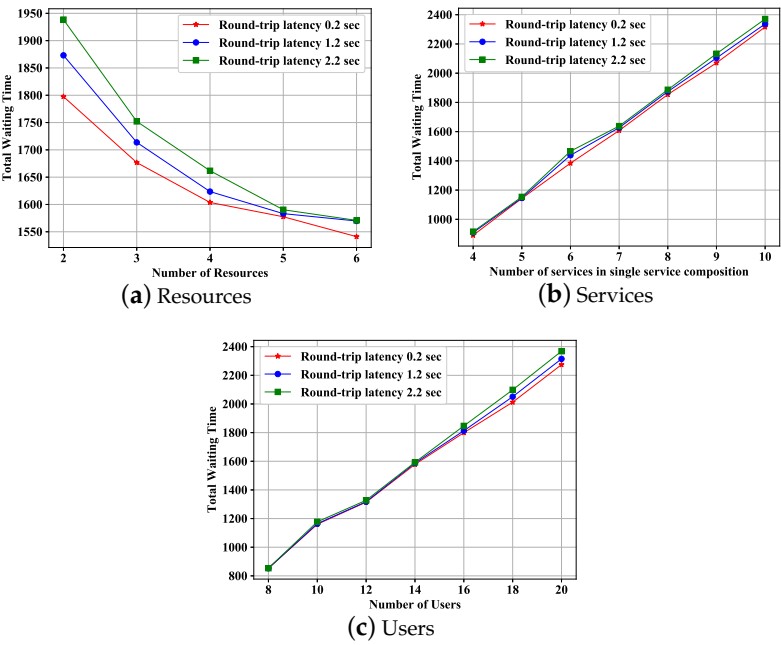

**Figure 7.** Impact of the round-trip on the total waiting time.

### 5.3.5. The Impact of Downtime

The impact of service downtime generated by service migration on total waiting time is shown in Figure 8. From the three figures, we can see that under the same condition, less service downtime will result in less total waiting time. In Figure 8a, the total waiting time decreased with increasing edge server resources. In this case, increasing the edge server's resources would enable more services to be deployed on edge servers. This means that there may be more service migration so that the impact of the size of the downtime latency on the total waiting time would be more obvious with more edge server resources. In Figure 8b,c, the more services in a single-service composition and more users resulted in more total waiting time. In Figure 8b, more services invoked by each user did not change the number of migrated services, so that the impact of downtime latency was not obvious. In

Figure 8c, with the increasing of users, more users were connected to the remote cloud servers. With the increasing of users' number, although the total waiting time increased, service migration would not happen, so that the impact of downtime latency was also not obvious.

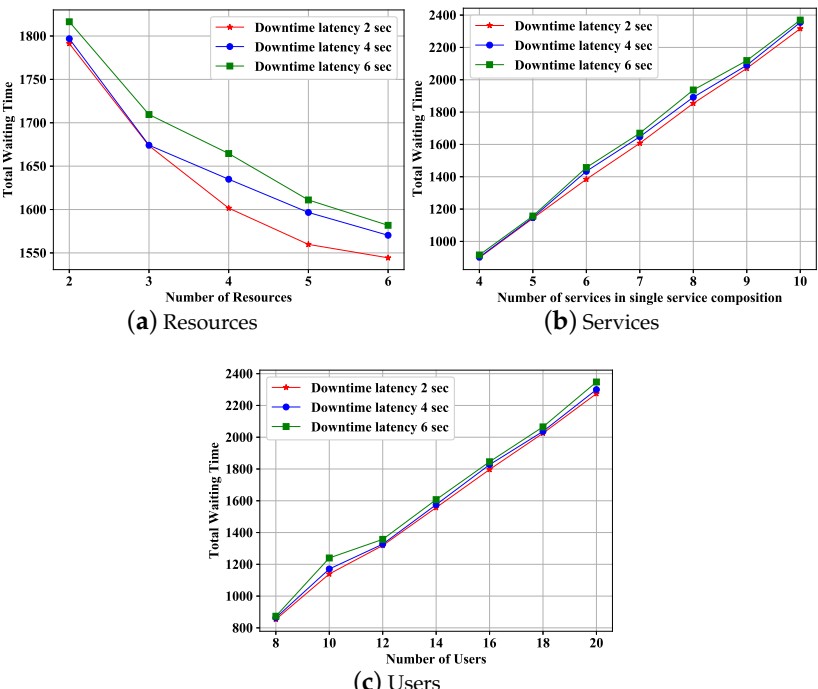

**Figure 8.** Impact of downtime on the total waiting time.

## 6. Related Work

The problem of multi-user edge server selection in mobile edge computing has been extensively investigated in the past few years in many research tracks [7,9,25–27]. These papers elaborated on how to select edge servers for multiple users from different aspects.

In [25], the authors studied the multi-user computation offloading problem for mobile-edge cloud computing in a multi-channel wireless interference environment. A multi-user computation offloading game was executed to decide whether the computing task was offloaded to an edge server. By the proposed approach, a Nash equilibrium was achieved, and the total offloading latency and energy consumption were minimized. The authors of [9] proposed how to dispatch and schedule the jobs in edge-cloud systems. They derived the first online job dispatching and scheduling algorithm, called *OnDisc*, which determines whether each job is to be processed locally at its device or offloaded to a server. The total weighted response time over all jobs was minimized by the proposed approach. Rather than rely on remote cloud servers, the multi-device task scheduling strategy for the ad hoc-based mobile edge computing system was proposed in [26]. The authors developed a multi-device distributed task scheduling game, which can make the task be offloaded to an optimal mobile device. When this game arrives at a Nash equilibrium possessing a finite improvement property, the overhead in term of time latency, energy consumption, and monetary cost is optimized. In [7], the authors modeled the user allocation problem as a bin packing problem, and the user allocation problem was solved as a series of connected integer linear programs. While satisfying the capacity constraint and proximity constraint, users were allocated to optimal hired edge servers to maximize the allocated number of users and minimize the hired number of edge servers. Considering the limited computing capacities and large amounts of peak load, a hierarchical edge cloud architecture was designed in [27] instead of using a flat collection of edge cloud servers. They developed workload placement algorithms that adaptively place users' workloads among different tiers of servers and decide how much computational capacity

is provisioned to execute each program, in order to make the average program execution latency minimized. Although the aforementioned papers achieved the goal of selecting optimal servers for many users, their research assumed that the users did not move. In real life, the users would move along a road over a period of time. Our research targeted much more realistic edge computing scenarios where the users move. At the same time, due to the limited coverage of a single edge server and the mobility of a user, this may lead to a significant decline in network performance or even disruption of ongoing mobile devices [5]. It would be difficult to ensure service continuity [28]. A preliminary work on mobility-driven service migration based on the Markov Decision Process (MDPs) was given in [29], which mainly considered one-dimensional (1D) mobility patterns with a specifically-defined cost function. Then, two-dimensions (2D) were proposed in [30] with distance-based MDP, which is a very realistic case compared to 1D mobility. They both computed a service migration policy to decide when the service migration was executed. In [5], the authors discussed the cutting-edge research efforts on service migration in MEC [28,31–33] and the process of service migration [34,35]. However, the authors only considered the service migration of a single user in a cellular system. Due to the limited capacity of an edge server, many users may influence each other when they select edge servers.

Our work tackled the multi-user edge server selection problem in scenarios with a known mobile path and services. We also realistically addressed this problem with respect to proximity constraints and capacity constraints. In this paper, the edge servers were selected in advance for multiple users to minimize the total waiting time.

## 7. Conclusions

In this paper, we focused on multi-user edge server selection in terms of minimizing the total waiting time of users in mobile edge computing. This problem was formally modeled, and we took the proximity constraint and capacity constraint into consideration to calculate the total waiting time. The MESP algorithm was adopted to achieve the objective of time-aware multi-user edge server selection. The simulation experiments showed that our MESP algorithm-based edge server selection method outperformed traditional methods in terms of time latency. However, the application of the proposed method was restricted. The mobile path and the speed of all mobile users should be precisely predicted. Moreover, the proposed algorithm selected edge servers only in terms of time consumption, and the energy consumption was not taken into consideration. In the future, we will select edge servers considering both time and energy consumption. We will also seek other algorithms to select the edge server for mobile users with higher effectiveness or efficiency.

**Author Contributions:** Conceptualization, W.Z. and Y.Z.; methodology, W.Z. and K.P.; software, Q.W.; validation, W.Z., Y.Z., and K.P.; formal analysis, Y.Z. and Q.W.; investigation, W.Z.; resources, W.Z.; data curation, Q.W.; writing, original draft preparation, W.Z. and K.P.; writing, review and editing, W.Z., Y.Z., Q.W., and K.P.; visualization, Y.Z.; supervision, Y.Z.; project administration, Y.Z.; funding acquisition, Y.Z.

**Funding:** This research was funded by the National Natural Science Foundation of China, Grant Number 61872002, the Anhui Key Research and Development Plan, Grant Number 201904a05020091, and the Natural Science Foundation of Anhui Province of China, Grant Number 1808085MF197.

**Conflicts of Interest:** The authors declare no conflict of interest.

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
