# Peer review of "Mobility-Enabled Edge Server Selection for Multi-User Composite Services"

_futureinternet, doi:10.3390/fi11090184_

Round 1

Reviewer 1 Report

The authors present multi-user edge server selection method based on particle swarm optimization. It is easy to follow the paper. It is suitable for both expert and non-expert readers. The paper is interesting. However, I have a few comments

1. The paper idea is not new as several works have discussed. What is the main difference between the proposed work and the state-of-the-art work

2. The introduction should be more elaborated. I suggest that MEC should be more explained and some of the motivations should be presented in the introduction section too

3. The paper is structured but it can be significantly improved

4. In the introduction section: What do you mean by "we proposed a new problem of selecting edge servers for many users in MEC....". 

5. In "Multi-user Edge Server Selection with Single Service", the authors explain how users access services when mobility occurs. In a point of view of users, they do not care about this. Edge services needs to take care of this. As we know that, Edge servers can be interconnected and communicate with each other. Therefore, they can share the information easily. In case of mobility, information of users or cookies can be exchanged really fast as Edge devices can be connected via Wi-Fi or Ethernet. 

There are a lot of methods which can be applied.

6. I somehow disagree with authors in Figure 3. I believe that each edge device or edge server is located in a geographical location. Therefore, an edge device or an edge server is able to serve all services. Could you please clarify this and why the architecture in Figure 3 is used instead of typical Edge architecture.

7. I suggest that the authors should compare the proposed method with other state-of-the-art methods. 

Reviewer 2 Report

In this work, authors present a novel method called MESP as multi-user edge server selection method based on swarm optimization. In this article, the proposed method is formally modeled to calculate the total waiting time of multi-user composite services. The manuscript needs to be significantly improved, and it needs more clarification in some places as follows:

- Contributions listed in introduction needs to be modified. Contributions to science are original methods and approaches that authors developed and contributed to the progress of the selected scientific area.

- I suggest authors to write some examples of edge resources to be exploited to run computing or storage services.

- Figure 1 is shown before it is referred in the text.

- In Figure 1, the horizontal axis is not defined. What "30kb/s" or similar represents for?

- In the text, it is written that "In this scenario, we assume that in path segment AB, user u1, u2, u3 want to invoke service T1, T2, T3". What are T1, T2 and T3? Why they are not shown in the figure?

- Authors have written "Considering the limited resources of each edge server whose 67 capacity is set as 4 unit of computing capacity, the workload of each service is set as 2 unit."

Does it mean, "3 services * 2 units = 6" that is bigger than 4 units of computing capacity? Therefore, what is the reason to have the workload more than the capacity of edge resources?

- In line 68, it is written that "Table 1, where the RR, UDS, DDS, RT and SD columns"

whereas there no column named RR in Table 1.

- In Figure 1, to cover the second situation, do we need to migrate services from one resource to another one? If yes, authors should explain the procedure in detail, and also the type of services. Or there is an instance of the service running on each edge resource? If so, what the column in Table 1 which shows the service downtime generated by service migration means?

- There are many big confusions in motivation scenarios. Scenarios are not clear, the figure is not self-explanatory, sentences are vague, explanation is not clear, etc.

- I see that authors mention a term called "known path". What is the difference between known and unknown paths?

- In the motivation Scenario, some parameters are introduced such as the requested resources, upload data size, download data size, the average service response time and the service downtime. Why these parameters are not considered in Section 3.1. Prerequisite Definitions?

- Authors need to mention what is the relation between the total waiting time and the response time of a service? This is because, the following sentence is written in the manuscript:

In this paper, we only consider one QoS property (response time). This is because users’ mobility only affects the variation of data transmission bandwidth between users and different edge servers, which would affect the total waiting time.

- About the previous point, can users’ mobility affect the reliability of the service? Can we consider the total waiting time as part of the service reliability?

- Here: "In the process of edge server selection, when the user is in a path segment, corresponding candidate edge servers can be selected."

Are we going to select a set of candidate edge servers or only one?

- Here: "The time consumption is mainly decomposed into three parts, including the time latency of transmitting input data, the time latency of transmitting output data and the response time. Sometimes, when the user connects to the remote cloud servers, the round-trip latency should be added to the time consumption"

What does transmitting input/output data mean?

Why round-trip latency is not part of transmitting time?

Why round-trip latency should not be calculated if the user connects to the edge node?

- It is written that "We assume that a random solution is generated for each user selecting servers and each user can select edge servers to invoke services depending on the random solution."

This sentence is different from what authors claimed before. They claimed that the selection algorithm is smart based on different necessary parameters. The reason to introduce the random algorithm is the comparison in the evaluation part?

- Authors mentioned some information about the evaluation setup: machine equipped with Inter Core i7 (3.6 GHz) and 16GB RAM. Could author let us know if there is a reason to select such machine?

- I wondered if authors to cite any possible research works using the random algorithm.

- What this sentence mean? The CPU capacity of each server is set within 2 - 6

- Why the number of users is set to 14? Why not for example 30?

- Why the number of services invoked by each user is randomly generated from 4 to 10? Why not for example from 10 to 50?

- And also about the number of edge server's resource? Moreover, what the number of edge server's resource means?

- The main drawback of the article is the results and evaluation. It needs to provide details about every single evaluation. The current form of the manuscript does not explain the diagrams at all, and the actual reason behind every one.

- I suggest authors to cite the following article for this sentence: "Therefore, the aggregate workload generated by services on a server must not exceed the remaining capacity of that server. At the same time, we assume that the remote cloud servers have sufficient computing resources and the user can connect them anytime from anywhere."

Taherizadeh, S., Stankovski, V. and Grobelnik, M., 2018. A Capillary Computing Architecture for Dynamic Internet of Things: Orchestration of Microservices from Edge Devices to Fog and Cloud Providers. Sensors, 18(9), 2938.

Minor comments:

we focus the problem --> we focus on the problem

T1, T2, T3 --> T1, T2 and T3

4 unit or 4 units, make sure

2 unit or 2 units, make sure

lager upload data size --> ?

an selected server --> a selected server

make the randomly random solution feasible --> ?

another servers --> another server

I hope authors find my comments useful for improving the paper. I wish all the best for them.

Round 2

Reviewer 1 Report

This version is better than the previous version. Although the authors' answers did not satisfy my comments completely, the answers are accepted. 

I believe that the paper is still in a border of acceptance and reject as the information provided in the manuscript is not completely persuasive. However, I see that the paper has some merits. I believe that this paper can draw a lot of readers.

It is suggested that some discussions should be added into the manuscript (i.e., in conclusion section).

Reviewer 2 Report

I appreciate the efforts made by authors for making changes for this revision, I think the quality of the manuscript has increased. Mainly, the questions in the previous review were asked from the reader point of view. I expect that authors would make the clarifications in the paper as well. In this way, the reader of the paper can follow the paper much easier and understand it better. I checked the answers, and found out that most of them are not included in the manuscript. Moreover, the following comments should be addressed completely:

- I see that authors wrote this sentence as contribution of the paper: "We consider how to select edge servers for mobile users in MEC.". I already mentioned that Contributions to science should be original methods and approaches that authors developed and contributed to the progress of the selected scientific area. Now, authors should ask themselves if considering how to select edge servers for mobile users in MEC is a contribution to science. The second contribution is also very detailed in the introduction that is not right.

- Again, I suggest authors to write at least an example of edge resource to be exploited to run computing or storage services. If I may understand correctly, authors wrote mobile device as example. Now, in section "5.2. Experiment Settings", authors wrote "The CPU capacity of each server is set within 2 - 6". I would like to make sure if there would be a consistency.

- Thanks author to explain what numbers mean in Figure 1. I asked this from a reader point of view. However, authors have not even defined in the figure yet. 

- Instance layer as part of service migration is not explained.

- Figure 1 is not yet self-explanatory enough. I suggest authors to significantly improve it.

- Explanation of "known path" in the author response letter is not put in the article.

- Could author make sure that this sentence written in the paper is correct: "users’ mobility only affects the variation of data transmission bandwidth". This is because the data-transmission bandwidth varies significantly depending on the chosen radio frequency and transmission power.

- I am not still convinced by the answer given for the following question:

Why round-trip latency is not part of transmitting time?

Why round-trip latency should not be calculated if the user connects to the edge node?

Because I see in the motivation example, it is mentioned that if the user moves from one geographic location to another one. Therefore, the round-trip time will be significantly changed.

I hope authors find my comments useful for improving the paper.

Round 3

Reviewer 2 Report

I appreciate the efforts made by authors for making changes for this revision, I think the quality of the manuscript has increased significantly now. More or less, my comments have been also addressed. The only comment is the necessity of improving Figure 2. Figure 2 is not self-explanatory. I suggest authors to revise this figure.
